# Injured adult motor and sensory axons regenerate into appropriate organotypic domains of neural progenitor grafts

Jennifer N. Dulin [1], Andrew F. Adler[1], Hiromi Kumamaru[1], Gunnar H.D. Poplawski[1], Corinne Lee-Kubli[1], Hans Strobl[1], Daniel Gibbs[1], Ken Kadoya[1,2], James W. Fawcett[3], Paul Lu[1,4] & Mark H. Tuszynski[1,4]

Neural progenitor cell (NPC) transplantation has high therapeutic potential in neurological disorders. Functional restoration may depend on the formation of reciprocal connections between host and graft. While it has been reported that axons extending out of neural grafts in the brain form contacts onto phenotypically appropriate host target regions, it is not known whether adult, injured host axons regenerating into NPC grafts also form appropriate connections. We report that spinal cord NPCs grafted into the injured adult rat spinal cord self-assemble organotypic, dorsal horn-like domains. These clusters are extensively innervated by regenerating adult host sensory axons and are avoided by corticospinal axons. Moreover, host axon regeneration into grafts increases significantly after enrichment with appropriate neuronal targets. Together, these findings demonstrate that injured adult axons retain the ability to recognize appropriate targets and avoid inappropriate targets within neural progenitor grafts, suggesting that restoration of complex circuitry after SCI may be achievable.

[1] Department of Neurosciences, University of California, San Diego, La Jolla, CA 92093, USA. [2] Department of Orthopaedic Surgery, Hokkaido University, Sapporo 060-8638, Japan. [3] John van Geest Centre for Brain Repair, Department of Clinical Neurosciences, University of Cambridge, Cambridge CB2 0SP, UK. [4] Veterans Administration Medical Center, San Diego, CA 92161, USA. Jennifer N. Dulin and Andrew F. Adler contributed equally to this work. Correspondence and requests for materials should be addressed to M.H.T. (email: mtuszynski@ucsd.edu)

Transplanted neural stem cells (NSCs) and neural progenitor cells (NPCs) can differentiate into mature neurons and functionally integrate into host circuitry, suggesting their potential therapeutic use for CNS trauma[1–3]. NSCs or NPCs derived from primary fetal tissue, embryonic stem cell (ESC) lines, or induced pluripotent stem cells (IPSCs) have been grafted into models of brain injury and spinal cord injury (SCI): in vivo, the cells consistently mature and extend large numbers of axons for long distances, forming functional synapses with host neurons[3]. In turn, host axons innervate grafts and form functional synapses onto graft-derived neurons[4–8]. In the context of CNS injury, such host-graft-host connectivity supports the establishment of new synaptic relays across lesion sites that enable significant functional improvement[7–10].

In recent years, accumulating evidence has demonstrated the ability of graft-derived neurons to spontaneously innervate their correct targets in the host CNS. For example, grafts of ESC-derived cortical cells to the intact and lesioned mouse cortex extend long-distance projections to subtype-appropriate host target regions[5,11,12]. Similarly, human ESC-derived dopaminergic neurons grafted to the substantia nigra, as well as embryonic brainstem-derived NSCs grafted into the injured spinal cord, preferentially innervate their correct targets in the host CNS, long distances from graft sites[13,14]. Importantly, the successful integration of graft-derived neurons into injured host circuitry appears to depend on the regional identities of grafted neurons; for example, graft-derived visual cortical neurons transplanted into the lesioned visual cortex integrate into injured host circuitry, whereas neurons with motor cortical identity fail to integrate into visual cortex[5]. These findings suggest that specific, functionally restricted connections from graft to host are spontaneously formed.

Axons of the injured host CNS also regenerate into grafts and form functional synapses[5,7–9,15–17]. However, it is not known whether lesioned adult axons that regenerate into grafts retain the capacity to innervate grafts in a target-specific manner. Previous work has indicated that injured adult axons can re-innervate their original host CNS targets if extrinsic manipulations (e.g., delivery of trophic factors) or intrinsic manipulations such as gene therapy to promote regrowth are employed[18–20]. In the context of neural cell transplantation into sites of spinal cord injury (SCI), we recently reported that injured axons of the adult corticospinal tract (CST) regenerate extensively into NPC grafts of caudal (spinal cord), but not rostral (forebrain), identity[7], indicating that homotypy of tissue grafts is critical to support regeneration of this injured motor projection. Rabies-mediated trans-synaptic connectivity mapping initiated from NPC grafts placed into spinal cord lesion sites demonstrated that graft neurons receive synaptic inputs from a diversity of host motor and sensory systems that normally innervate spinal cord targets[8]. However, it remains unknown whether new connections from host to graft are established with appropriate or inappropriate graft-derived neurons.

We now report that spinal cord NPCs transplanted into the injured adult spinal cord differentiate into multiple subtypes of neurons with distinct lineage identities, and that the proportions of these subpopulations can be manipulated by selective transplantation of regionally-restricted progenitors. We show that dissociated NPC grafts self-assemble neuronal clusters with laminar organization of distinct dorsal spinal cord interneuronal subtypes, recapitulating the laminar cytoarchitecture of the intact spinal cord dorsal horn. Remarkably, these organotypic dorsal horn-like structures are preferentially innervated by regenerating host sensory afferent axons, but are entirely avoided by regenerating host corticospinal axons. Finally, we demonstrate that the extent of regeneration of specific host axons is significantly enhanced in grafts of dorsal-restricted spinal cord NPCs compared to ventral spinal cord NPC grafts. These findings demonstrate that dissociated NPCs grafted into the injured CNS environment can partially recapitulate the organization of the adult spinal cord dorsal horn. Notably, injured, regenerating adult axons retain their developmental ability to distinguish between functionally appropriate and inappropriate targets within NPC grafts.

## Results

**Spinal cord NPCs self-assemble dorsal horn-like domains.** The intact spinal cord contains hundreds of phenotypically and functionally distinct subpopulations of neurons[21–23], and their integration into specific functional circuits is critical for the faithful transmission of motor and sensory information[23,24]. We and others have previously reported that human and rodent spinal cord NPCs differentiate into mature neurons and glia following engraftment into the injured adult spinal cord[7–9,15,16]. However, the phenotypic fates of graft-derived neurons have not been well characterized. During spinal cord development, patterning of the neural tube leads to the specification of discrete progenitor domains along the dorsal/ventral axis that give rise to thirteen cardinal neuronal lineages (Supplementary Fig. 1a)[23]. For the present study, as well as previous transplantation studies utilizing dissociated rodent spinal cord NPCs, donor cells are obtained from E14 rat embryos[7,15,25–30], at which stage the spinal cord contains a mix of NSCs, progenitors and post-mitotic cells[28,31]. To characterize the types of neural progenitors that are present in the cell suspension used for grafting, we dissociated E14 spinal cords using previously described methods[26], cultured cells for 24 h, and characterized expression of NPC-specific transcription factors (Fig. 1a–e). After 1 day in vitro (DIV), approximately 25% of cells expressed Sox2, a marker of NSCs and neuronal progenitors[32] (Fig. 1a, e). We also observed expression of A2B5, a marker of glial-restricted precursor cells[33] (Fig. 1b). NPCs expressed transcription factors specific to dorsal (Pax7), as well as ventral (Pax6, Nkx6.1) progenitors (Fig. 1c–e)[34–39], indicating that both dorsally-fated and ventrally-fated spinal cord NPC populations are present in the initial cell preparation used for grafting.

For the present study, we focused our characterization on graft-derived neurons with dorsal spinal cord identities because post-mitotic dorsal horn neurons express unique combinations of lineage-specific and lamina-specific molecular markers[23,40] that enable these cell types to be readily identified (Supplementary Fig. 1b). We cultured E14 cells for 10 DIV and examined expression of multiple post-mitotic dorsal horn neuronal markers. At this stage, large clusters of cells were visible (Supplementary Fig. 1c). Cells within these clusters expressed multiple post-mitotic dorsal horn markers including: Lbx1, a transcription factor marker of dI4-dI6 neurons[41–44]; Tlx3, a transcription factor expressed in excitatory dI5/dIL$_B$ neurons[45–48]; and calretinin, a calcium-binding protein expressed by both excitatory and inhibitory neurons in the superficial dorsal horn[49,50] (Fig. 1f–j). These cultures also included ventral spinal cord neuronal subtypes; for example, we identified neurons immunoreactive for Foxp2, a transcription factor expressed in V1 ventral spinal cord interneurons[51], as well as dI2-derived neurons that settle in the ventral spinal cord (Fig. 1i)[52]. At 10 DIV, Lbx1$^+$ and Tlx3$^+$ neurons comprised approximately 15 and 13% of total cells, respectively (Fig. 1j). Notably, neurons with dorsal spinal cord identities were located within the same large cell clusters (Fig. 1k), whereas ventral interneuron types were primarily located outside of these clusters at 10 DIV (Fig. 1l).

To characterize cell fates after grafting in vivo, we transplanted dissociated spinal cord NPCs into sites of cervical spinal cord

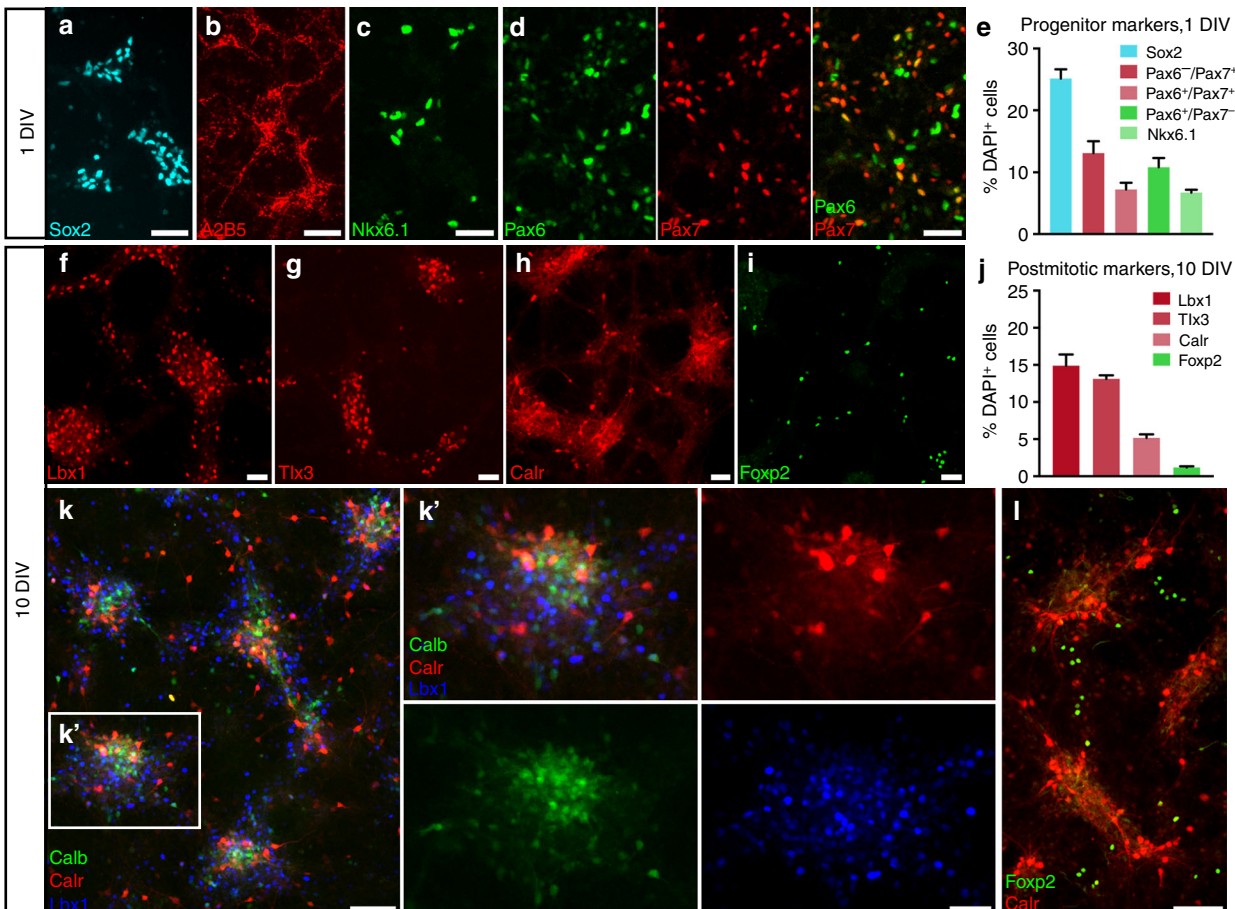

**Fig. 1** Dissociated spinal cord NPCs differentiate into dorsal and ventral neuronal lineages in vitro. Spinal cords from E14 rat embryos were dissociated, and cells were cultured for either (**a–e**) 24 h or (**f–l**) 10 days. **a** Sox2+ neuronal progenitors and **b** A2B5+ glial progenitors are both present at 1 day in vitro (DIV). **c** Nkx6.1+, **d** Pax6+ (green) and Pax7+ (red) progenitors are present at 1 DIV. Pax6 is expressed in both dorsal and ventral progenitors, but all Pax7+ cells are dorsally-fated. **e** Quantification of percent of total cells expressing NPC markers at 1 DIV; mean + SEM based on total cell counts for 3 independent biological replicates. **f** Lbx1+, **g** Tlx3+, **h** calretinin+ (Calr), and **i** Foxp2+ cells are present at 10 DIV. **j** Quantification of percent of total cells expressing markers of dorsal (Lbx1, Tlx3, calretinin) and ventral (Foxp2) post-mitotic neurons at 10 DIV; mean + SEM based on total cell counts for 4 different biological replicates. **k** Neurons expressing dorsal horn markers calretinin (red), calbindin (green), and Lbx1 (blue) are located within the same cell clusters after 10 days in culture. **k'** High-magnification view of boxed area in (**k**) demonstrates clustering of multiple dorsal interneuron subtypes. **l** In contrast, Foxp2+ (green) ventral interneurons are not located within dorsal interneuron clusters. Scale bars = 100 μm (**k**, **l**), 50 μm (**a–d**, **f–i**, **k'**)

dorsal column injury in adult rats (Fig. 2a), and examined grafts at six weeks following transplantation. At this time point, grafts were densely populated with mature neurons (Fig. 2b), with 34,900 ± 1500 NeuN+ cells/mm³ of graft tissue (mean ± SEM, n = 9). In spinal cords of intact rats that were developmentally age-matched to grafts (postnatal day 35), normal spatial distributions of dorsal horn neurons expressing laminae I–IV markers were apparent (Fig. 2c, d). Neurons expressing dorsal horn markers were also abundant in differentiated NPC grafts (Fig. 2e, f); however, despite the initial transplantation of a dissociated mix of progenitors, dorsal spinal cord cell types were not evenly interspersed throughout grafts as might be expected. Rather, cells were confined to discrete clusters (Fig. 2e, f), evocative of the clusters formed after 10 days in culture (Fig. 1k, l), as well as the dense, clustered bands within the intact dorsal horn (Fig. 2c, d). In addition to the clustered calretinin+ neurons, we also detected non-clustered, larger-diameter calretinin+ neurons dispersed throughout grafts; this is similar to the intact spinal cord, in which smaller-diameter calretinin+ neurons in the superficial laminae are densely clustered, whereas larger-diameter calretinin+ neurons in the ventral and intermediate gray matter are more widely dispersed and not clustered together[25] (Supplementary Fig. 2a–b).

A defining feature of the spinal cord dorsal horn is its laminar distribution of molecularly distinct neuronal subtypes[23]. For example, dorsal horn neurons expressing calretinin and/or Tlx3 are largely confined to the same spatial domain in laminae I-II, whereas Lbx1+ neurons primarily reside in the deeper dorsal horn laminae III–IV; additionally, a few neurons in the interstitial zone between laminae II and III express both markers (Supplementary Fig. 2c–e). In differentiated grafts, we identified similar distributions of laminae-specific cell subtypes; for instance, calretinin+ neurons and Tlx3+ neurons, which normally reside in the same laminae I-II band (Supplementary Fig. 2c), likewise populated the same regional domains of grafts (Supplementary Fig. 2f). Furthermore, Lbx1+ neuron clusters in grafts were located adjacent to Tlx3+/calretinin+ clusters, with some cells at the border of these zones expressing both markers (Supplementary Fig. 2g–h). Because this organization of neurons suggested the preservation of "laminae" in grafts, we assessed whether more complex laminar distribution of multiple neuronal subtypes was evident. Indeed, probing grafts with different combinations of laminae I-IV markers confirmed the correct laminar arrangement of multiple distinct dorsal horn neuronal subtypes (Fig. 2g, h, Supplementary Fig. 3a–b). Hence, the organization of these multicellular domains correctly recapitulates the layered

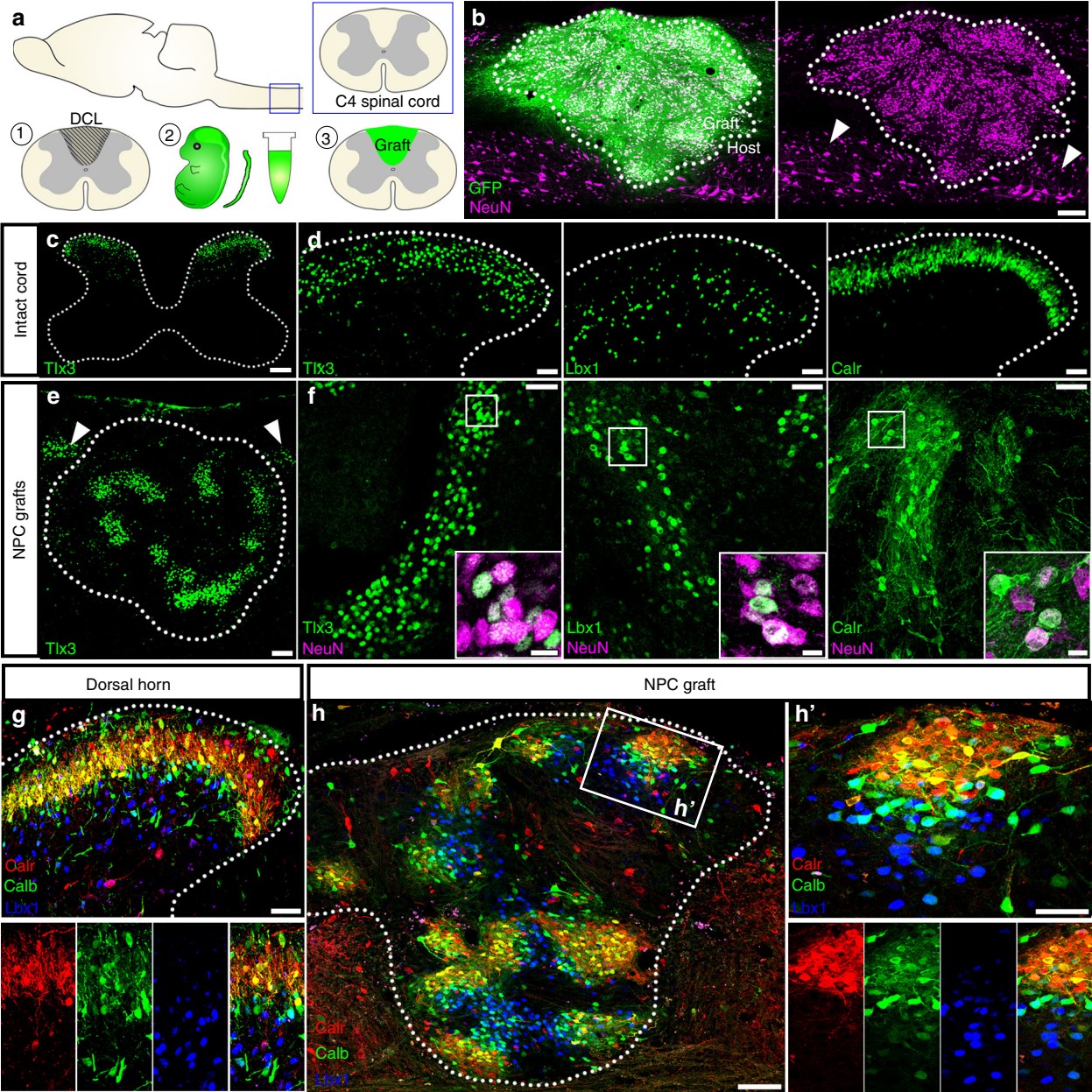

**Fig. 2** Dorsal spinal cord interneurons form laminar, dorsal horn-like structures within NPC grafts. **a** Cartoon schematic illustrating experimental paradigm. Following placement of a dorsal column lesion at spinal cord level C4 in adult rats, spinal cords from E14 rat embryos were isolated and dissociated into a single-cell suspension. Cells were grafted immediately into sites of spinal cord injury, and grafts were allowed to mature. Panels **b**, **e**, **f**, and **h** are images of spinal cord NPC grafts at six weeks post-transplantation; panels **c**, **d**, and **g** are images of cervical spinal cord from intact, age-matched rats (postnatal day 35). **b** Sagittal view of GFP+ (green) NPC graft at six weeks post-transplantation. Graft (dotted lines) is populated with NeuN+ (magenta) neurons; host spinal cord neurons are indicated with arrowheads. **c** Transverse image of intact cervical spinal cord demonstrating that interneurons expressing Tlx3 (green) are located within the superficial dorsal horn. **d** High-magnification images of intact cervical dorsal horn showing distributions of neurons expressing dorsal horn markers Tlx3 (left), Lbx1 (middle), and calretinin (Calr; right). **e** Low-magnification sagittal image of NPC graft with clustering of Tlx3+ neurons. Dorsal band of host Tlx3+ neurons is also visible (arrowheads). **f** Images from NPC grafts showing clusters of neurons immunoreactive for Tlx3 (left), Lbx1 (middle), and calretinin (right). Insets show co-localization of markers with NeuN (magenta). **g**, **h** Laminar distribution of cells immunoreactive for laminae I–II interneuron markers calretinin (red) and calbindin (Calb, green), and the laminae III–IV interneuron marker Lbx1 (blue) in (**g**) the intact spinal cord dorsal horn and (**h**) NPC graft. **h** Grafts contain multiple, layered clusters of dorsal interneurons; (**h'**) high-magnification view of a dorsal horn-like cluster in graft. Scale bars = 200 μm (**b**, **c**), 100 μm (**e**, **h**), 50 μm (**d**, **f**, **g**, **h'**), 10 μm (insets in **f**)

cytoarchitecture of the intact dorsal horn. Every graft examined contained multiple dorsal horn-like clusters (Fig. 2h), and the neurons within these clusters were always GFP+ (Supplementary Fig. 3c). Although these domains were not oriented in any particular direction with respect to the host anatomical axes, neurons within individual clusters were arranged in such a way that the clusters had clear "dorsal" and "ventral" aspects. Additionally, these clusters sometimes had a curved or arc shape that was particularly evident in laminae I–II regions, resembling the natural arc of the intact dorsal horn (Supplementary Fig. 3d–e).

**Graft-derived sensory domains are innervated by CGRP$^+$ axons**. We have previously reported that multiple types of injured host axons spontaneously regenerate into NPC grafts and form functional synaptic connections with graft-derived neurons[7–9]. However, it remains unknown whether new host-to-graft connections are established onto functionally appropriate targets. The presence of discrete dorsal interneuron domains in mature NPC grafts presented a unique opportunity to address this question. Spinal cord laminae I–II contain a heterogeneous mixture of neuronal subtypes, many of which are integrated into pain-processing circuits and are innervated by small diameter nociceptive afferents such as peptidergic C-fibers expressing calcitonin gene-related peptide (CGRP)[23,45,53] (Fig. 3a, b). We therefore investigated whether laminae I-II domains in NPC grafts receive preferential inputs from host CGRP$^+$ axons. Indeed,

we found that host CGRP$^+$ afferents penetrated NPC grafts and underwent extensive arborization within these domains (Fig. 3c). CGRP$^+$ axon density within NPC grafts was significantly enriched in calretinin$^+$ clusters ($p = 0.01$; Supplementary Fig. 4b), reflecting the significant enrichment of these axons within the superficial laminae of the intact spinal cord ($p = 0.001$; Supplementary Fig. 4a). Calretinin$^+$ cell clusters located more dorsally within grafts were in closest proximity to incoming host sensory axons, and were more densely penetrated by host CGRP$^+$ axons than calretinin$^+$ cell clusters located more ventrally and distantly within grafts (Fig. 3c).

We recently reported that CGRP$^+$ neurons in the host dorsal root ganglia establish functional synaptic connections with NPC graft-derived neurons, as identified using modified rabies-mediated trans-synaptic connectivity mapping[8]. To determine

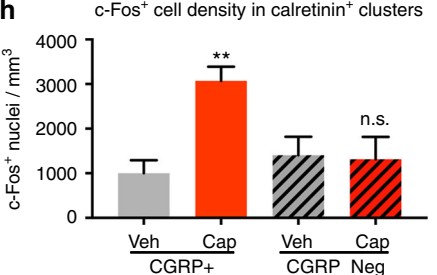

whether these synaptic contacts are made onto appropriate target domains in grafts, we examined neurotransmission of afferent sensory information. In the intact system, noxious stimulation of the peripheral nerves via subcutaneous capsaicin injection activates capsaicin-sensitive DRG neurons, resulting in the activation of their central axon terminals in the dorsal horn[54–58] and their targets in dorsal horn laminae I-II, including a subset of calretinin[+] neurons[59]. Ninety minutes following subcutaneous injection of capsaicin into the forelimbs of uninjured adult rats, we observed a 5-fold increase in the number of cells expressing the neuronal activation marker c-Fos within laminae I-II of the cervical spinal cord, compared to vehicle-treated animals ($p = 0.001$, Fig. 3d, e). Robust c-Fos immunoreactivity was observed in cells including calretinin[+] neurons in these laminae (Fig. 3d). We next investigated whether subcutaneous capsaicin injection could also activate neurons within laminae I–II domains of NPC grafts within sites of cervical SCI. In both vehicle-treated and capsaicin-treated animals, c-Fos[+] nuclei were distributed throughout graft tissue, indicating a baseline level of graft neuronal activity in anesthetized animals (Fig. 3f, g). Total numbers of c-Fos[+] nuclei in all regions of graft tissue (not separated according to calretinin[+] clusters) revealed no significant difference between groups (vehicle = $1850 \pm 346$ nuclei/mm$^3$, capsaicin = $2140 \pm 208$ nuclei/mm$^3$, mean $\pm$ SEM; $n = 3$ per group; $p = 0.51$ by two-tailed t-test). However, when comparing graft-derived calretinin[+] cell domains that were penetrated by CGRP[+] axons, treatment with capsaicin resulted in a significant, 3-fold increase in the number of c-Fos-expressing neurons compared to vehicle-injected controls ($p = 0.008$, Fig. 3h). In contrast, calretinin cell clusters in grafts lacking innervation by CGRP[+] axons exhibited no response to capsaicin injection (Fig. 3h), confirming that host axonal inputs into calretinin[+] cell clusters were required to elicit c-Fos activation. These findings provide evidence that host nociceptive sensory axons support the functional transmission of noxious stimuli to post-synaptic targets within appropriate functional domains of NPC grafts.

**Host corticospinal axons avoid graft sensory clusters.** We next examined whether the regeneration of other host axon projections into grafts is also influenced by graft topography. Specifically, we assessed whether graft-derived laminae I–II clusters are avoided by regenerating host axons that are inappropriate presynaptic partners for dorsal sensory neurons. We focused our characterization on the corticospinal tract, because these axons regenerate robustly into spinal cord NPC grafts[7] yet laminae I–II

sensory interneurons constitute functionally inappropriate targets for this descending motor projection. In the intact cervical spinal cord, density of anterogradely-labeled corticospinal axons was highest in the deeper dorsal laminae and in the intermediate gray matter; in contrast, axon density was relatively low in motor pools (lamina IX) and in laminae I–II, the superficial laminae of the dorsal horn (Fig. 4a–c). Despite a high density of CST axons within laminae III–IV of the dorsal horn, the immediately adjacent and more superficial laminae I–II were depleted of corticospinal axons. In contrast to the substantial enrichment of CGRP[+] axons in the calretinin[+] band of the dorsal horn, CST axon density was substantially attenuated in laminae I–II ($p = 0.015$, Supplementary Fig. 4a, Fig. 4d, e).

We then examined the topographic specificity of corticospinal axon regeneration into NPC grafts. Corticospinal axons regenerated extensively into grafts, consistent with previous observations[7]. However, the density of these axons was consistently and significantly attenuated in graft laminae I-II domains ($p < 0.0001$, Supplementary Fig. 4b, Fig. 4f). In stark contrast to the preferential innervation of these domains by host CGRP[+] axons, they were consistently avoided by regenerating CST axons. Indeed, the presence of a calretinin[+] cluster adjacent to the rostral graft/host interface was associated with a failure of corticospinal axons to regenerate into grafts at these sites (Fig. 4g). Together, the contrasting preference/avoidance of these distinct host axon systems with respect to laminae I–II graft domains recapitulates their normal termination patterns in the intact system. Collectively, these results show that the distribution of organotypic domains in NPC grafts has a potent effect on the extent and patterns of innervation by distinct motor and sensory host axon systems. Moreover, this more broadly demonstrates the capacity of injured adult axons to exhibit topographically- and developmentally-appropriate regeneration into newly placed neural grafts, with preferential innervation of correct graft-derived targets and avoidance of incorrect targets.

In a time course study, we have previously determined that corticospinal axon regeneration into spinal cord NPC grafts is initiated approximately 10–14 days after transplantation. This raises the question whether the initial period of host axon regeneration is influenced by graft topography. We hypothesized that early-stage (~2 week post-transplantation) grafts would contain organized dorsal interneuron domains based on our observations that: (**1**) dissociated NPCs form clusters of dorsal interneurons after 10 days in culture (Fig. 1k), that (**2**) distinct interneuron bands are apparent in the dorsal horn of rats at

**Fig. 3** Graft-derived laminae I-II domains are preferentially innervated by regenerating host CGRP[+] axons. Images of (**a**, **b**, **d**) cervical spinal cord from intact, age-matched rats and (**c**, **f–h**) spinal cord NPC grafts at 6 weeks post-transplantation. **a** Primary afferent CGRP[+] axons (cyan) terminate within the superficial dorsal horn of the intact C4 spinal cord. **b** Termination of CGRP[+] axons (cyan) overlaps substantially with the laminae I–II band of calretinin (Calr, red) immunoreactivity in the intact C4 dorsal horn. Dotted line indicates border of calretinin[+] immunoreactivity. **c** Host CGRP[+] fibers (cyan) penetrate spinal cord NPC grafts and densely innervate graft-derived calretinin[+] clusters (red). Graft/host border indicated with dotted line. **d** Sagittal images of cervical dorsal horn of uninjured animals sacrificed 90 min after subcutaneous injection of either vehicle (VEH) or capsaicin (CAP) into the forearms. c-Fos immunoreactivity is abundant in nuclei of cells within the laminae I–II calretinin[+] band (red) in capsaicin-treated animals (CGRP, blue). Inset: High-magnification image of dorsal horn laminae I–II in capsaicin-treated animals reveals c-Fos immunoreactivity (green) in nuclei of calretinin[+] neurons (red; arrowhead); CGRP, blue. **e** Quantification of the number of c-Fos[+] nuclei per mm$^3$ volume within the laminae I-II calretinin[+] band of intact cervical spinal cord following 90 min treatment with either vehicle ($n = 3$) or capsaicin ($n = 3$). Mean $\pm$ SEM; ***$p < 0.001$ by two-tailed t-test. **f**, **g** Images of NPC grafts from animals receiving subcutaneous injections into the forelimbs of either (**f**) vehicle or (**g**) capsaicin 90 min prior to sacrifice, showing c-Fos (green), calretinin (red), and CGRP (blue) immunoreactivity. **f′**, **g′** High-magnification images of grafts. **f′** In a graft from a vehicle-treated animal, a few c-Fos[+] nuclei (green) are present within a calretinin[+] neuron (red) patch heavily innervated by host CGRP[+] axons (blue). **g′** In a graft from a capsaicin-treated animal, many c-Fos[+] nuclei (green) are present within a calretinin[+] neuron (red) patch with modest innervation by host CGRP[+] axons (blue). **h** Quantification of the number of c-Fos[+] nuclei per mm$^3$ volume within calretinin[+] graft domains with CGRP[+] axon innervation (solid), and calretinin[+] graft domains without CGRP[+] axon innervation (diagonal stripes) at 90 min after subcutaneous injection with either vehicle (gray; $n = 3$) or capsaicin (red; $n = 3$). Mean $\pm$ SEM; **$p < 0.01$ by two-tailed t-test (Veh vs. Cap treatment was compared for either CGRP[+] or CGRP-negative clusters). Scale bars = 100 μm (**a**, **c**, **d**, **f**, **g**), 50 μm (**b**, **c′**, **f′**, **g′**), 10 μm (inset in **d**)

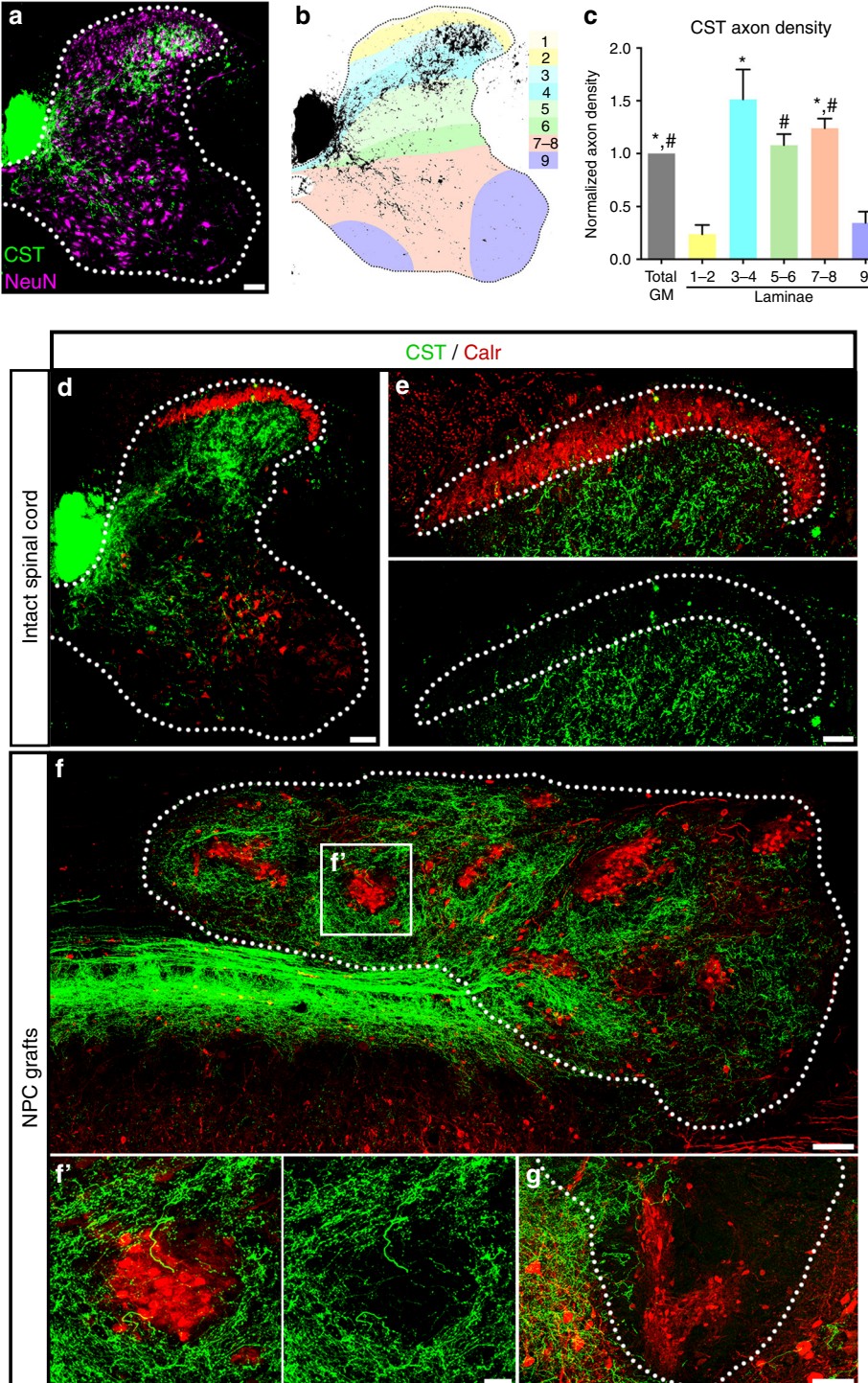

**Fig. 4** Graft-derived laminae I–II domains are avoided by host corticospinal axons. **a** Representative image of corticospinal tract (CST) axon (green) termination within the intact cervical spinal cord of age P35 rat; neurons labeled with NeuN (magenta). Border of spinal cord gray matter indicated with dotted lines. **b** Cartoon illustrating distribution of corticospinal axons (black) against map of Rexed laminae I–IX. **c** Quantification of CST axon density within laminae I–IX; data are expressed as mean ± SEM and normalized to CST axon density in total gray matter (GM); *$p < 0.05$ vs. laminae I–II and #$p < 0.05$ vs. lamina IX by one-way ANOVA followed by Tukey's post-test for multiple comparisons; $n = 4$. **d**, **e** Labeling of CST axons (green) and calretinin (red) in the intact C4 spinal cord. **e** High-magnification image of CST axons in the intact dorsal horn. Dotted lines in **e** indicate border of calretinin immunoreactivity. **f** CST axons (green) regenerate extensively into spinal cord NPC graft (dotted line). CST axons are attenuated within regions of graft tissue with calretinin+ domains (red). **g** Calretinin+ domain (red) near the rostral graft/host interface appears to block deeper penetration of CST axons (green) into graft (dotted line). Scale bars = 100 μm (**a**, **d**, **f**), 50 μm (**e**, **g**), 25 μm (**f'**)

postnatal day 5 (Supplementary Fig. 5a–b). Indeed, we detected clusters of graft-derived dorsal horn interneurons two weeks post-transplantation (Supplementary Fig. 5c–d). Moreover, regenerating CST axons avoided these regions at this time point (Supplementary Fig. 5e) and CGRP$^+$ axon innervation of clusters was already evident (Supplementary Fig. 5f). These results indicate that laminae I–II neuron clusters are formed early after grafting and are present at the initial time period of host axon innervation of grafts. Thus, these domains could act as blockades to actively regenerating corticospinal axons, rather than developing later after the bulk of CST regeneration has already occurred.

**Dorsalized grafts support greater host axon regeneration.** Our results thus far demonstrate that the distribution of regionally-specific neuronal subtypes in NPC grafts has a strong influence on host axon innervation. We hypothesized that by grafting regionally-distinct spinal cord progenitor populations, the distribution of neuronal subtypes in differentiated grafts, and in turn the extent of host axon regeneration into grafts, could therefore be modulated. During normal spinal cord development, thirteen progenitor domains are specified along the dorsoventral axis of the neural tube, and each class of progenitor gives rise to distinct classes of dorsal and ventral neurons (Supplementary Fig. 1a)[23]. The pdIL progenitor domain generates the late-born dorsal interneuron (dIL) class of neurons, which include all of the subtypes populating laminae I–III of the dorsal horn[40]. Early post-mitotic dIL neurons express Lbx1 and are restricted to the dorsal half of the cord by E12 in mouse[44,60] and E14 in rat (Fig. 5a). We therefore took advantage of the spatial segregation of dorsal and ventral interneuron progenitors in rats at E14, the stage at which we obtain spinal cord NPCs for transplantation. We isolated NPCs from either the dorsal or ventral half of microdissected rat E14 spinal cords for transplantation into sites of SCI (Fig. 5b). Six weeks later, we assessed graft distribution of dorsal interneuron markers, as well as the spinal cord V1a interneuron marker Foxp2, which is restricted to neurons of the ventral spinal cord during both embryonic[61] and postnatal stages[51] (Fig. 5a, Supplementary Fig. 6a). This approach was highly effective, as ventral NPC grafts were significantly enriched for Foxp2$^+$ ventral interneurons and depleted of dorsal Tlx3$^+$ and Lbx1$^+$ interneurons at six weeks post-transplantation, and the inverse was true for dorsal NPC grafts (Fig. 5c, Supplementary Fig. 6c–f). Interestingly, although each examined subtype of dorsal interneuron was significantly depleted in ventral NPC grafts, we did not detect any significant differences in the relative proportions of these markers between whole and dorsal NPC grafts (Supplementary Fig. 6d–f). This observation may reflect the increased proliferative capacity and/or survival of dorsal NPCs compared to ventral NPCs, resulting in a higher ratio of dorsal versus ventral neurons in differentiated grafts compared to the starting amount of NPCs grafted. This possibility is supported by the observation that dorsal NPC grafts were significantly larger than ventral grafts, despite grafting equal numbers of cells (Supplementary Fig. 6b).

Having confirmed that dorsal and ventral neuron populations in grafts were effectively modified by this dissection approach, we next assessed the extent of host axon regeneration into each graft type. CGRP$^+$ axons terminate strictly in the dorsal spinal cord (Fig. 3a), so we hypothesized that regeneration of these axons would be enhanced in dorsal NPC grafts. Indeed, CGRP$^+$ axon density normalized to total graft size was significantly higher in dorsal grafts than in ventral grafts (Fig. 5d, e), demonstrating that selective transplantation of dorsal spinal cord progenitors, including dorsal horn progenitors, effectively enhances graft

CGRP$^+$ axon innervation by enriching for their primary target cell population. Interestingly, we found that corticospinal axon density was also significantly greater in dorsal grafts compared to ventral grafts, (Fig. 5f, g), an observation consistent with the fact that the majority of anterogradely-labeled corticospinal axons terminate in dorsal laminae III–VI of the intact rat spinal cord (Fig. 4a–c). Together, these findings demonstrate that manipulating the cellular composition of NPC grafts significantly influences the extent of host sensory and motor axon regeneration into grafts.

## Discussion

Understanding the interactions between graft-derived neurons and host CNS circuitry will be key in the optimization of cell transplantation approaches for neurological disease and injury. Previous studies have demonstrated the intrinsic ability of graft-derived neurons to innervate appropriate subtype-specific targets in the host CNS[5,12,13], and also within the grafts themselves: for example, grafted primate serotonergic neurons densely innervate grafted dopaminergic neurons within the same mesencephalic transplant, mirroring normal projection patterns of the intact system[62]. However, the appropriateness of host innervation of neural grafts has not been addressed previously, yet is a critically important issue in efforts to restore connectivity of host systems across spinal cord injury sites. Our results demonstrate that injured host motor and sensory axons selectively innervate graft tissue domains populated with specific neuronal subtypes and avoid domains populated with inappropriate targets. Importantly, this occurs spontaneously and without the need for additional manipulations to promote target-directed guidance, e.g., exogenous delivery of chemotrophic gradients. This is a promising finding in the context of CNS trauma, in which major therapeutic goals include not only regeneration of host axons, but promoting their integration into new, functionally relevant neural circuits. It will be important for future work to determine the extent to which graft-derived neurons integrate into host neural circuitry, toward the ultimate goal of restoring lost function.

The potential of the injured host CNS environment to influence fates and/or organization of grafted cells is an important consideration for cell transplantation studies. In the current study, several lines of evidence reduce the probability that interactions with the host environment influence either the identities of graft-derived neurons or their organization into organotypic multicellular domains. First, the interneuron identities observed in mature grafts were also adopted by NPCs cultured in vitro (Fig. 1f–l). This indicates that NPCs isolated from the embryonic spinal cord are already committed to specific developmental fates, and that transplantation into adult lesion sites is not necessary for acquisition of these fates. Moreover, cells with dorsal identities were located within the same clusters in culture (Fig. 1k), illustrating the spontaneous organization of "dorsal" interneuronal domains in the absence of an in vivo host spinal cord environment. Finally, within GFP$^+$ NPC grafts we never observed GFP-negative neurons (Supplementary Fig. 3c), indicating their graft cell origin and ruling out the possibility that these structures might be comprised of host neurons migrating into grafts. Thus, it appears unlikely that interactions with the host spinal cord influence the formation of organotypic graft-derived structures. Rather, our results indicate that the acquisition of specific fates by NPCs and their ability to assemble organized multicellular domains are intrinsic properties of graft-derived cells.

Jakeman and colleagues previously reported that rat fetal spinal cord transplanted into the adult CNS developed substantia gelatinosa-like patches[63], illustrating the ability of the embryonic

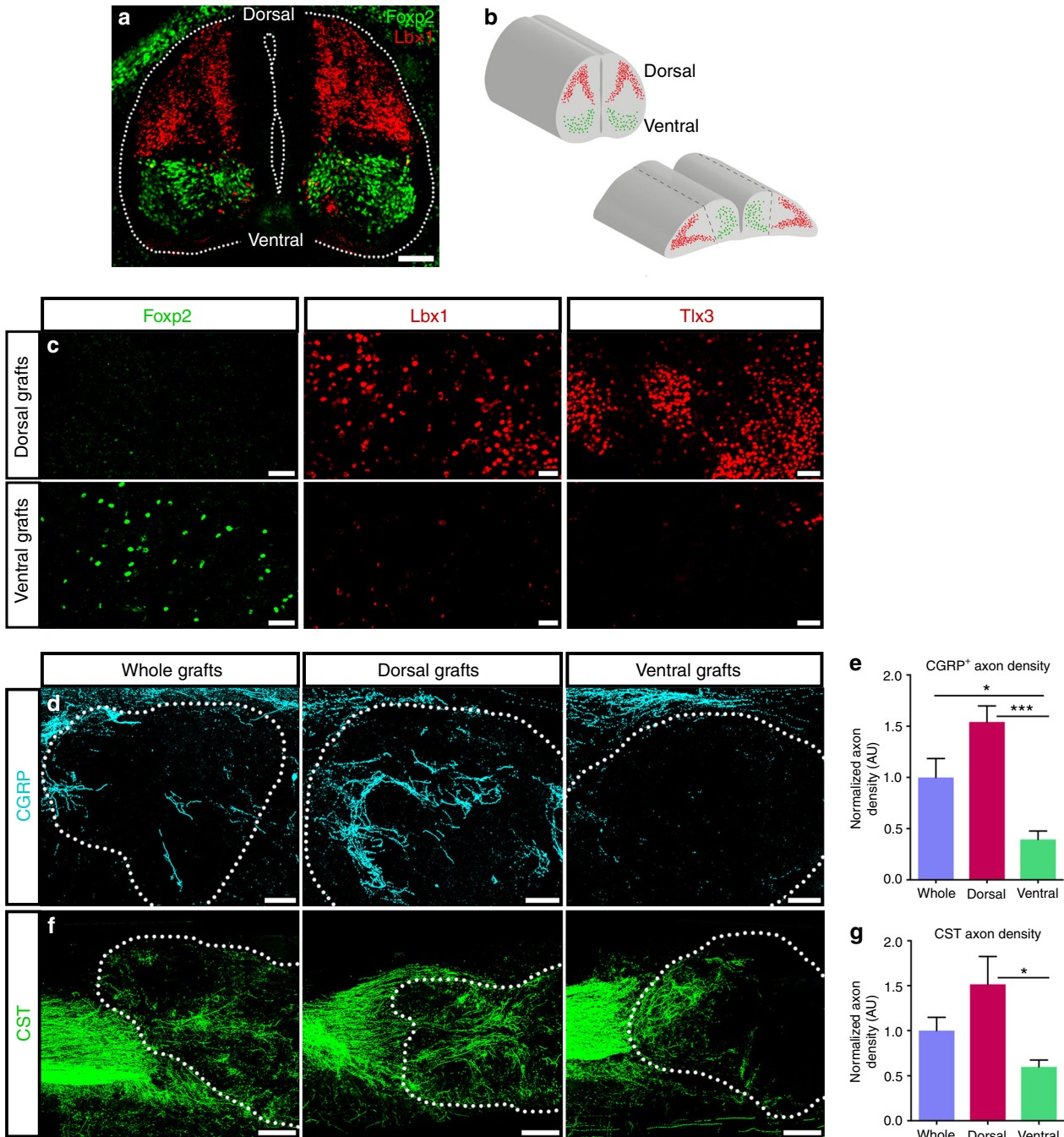

**Fig. 5** Host corticospinal and CGRP[+] axons preferentially innervate dorsal spinal cord NPC grafts. **a** Transverse image of E14 spinal cord. Lbx1 (red) is expressed in post-mitotic dIL neurons in the dorsal half of the spinal cord, and Foxp2 (green) is expressed in V1 neurons in the ventral half of the spinal cord. **b** Cartoon illustrating isolation of dorsal and ventral spinal cord NPCs. Top: whole E14 rat spinal cords containing dorsal (red) and ventral (green) NPCs. Bottom: "open-book" E14 spinal cords were separated along the dorsal/ventral boundary (dashed lines) and dorsal or ventral halves of spinal cord were dissociated for grafting. **c** Representative images from dorsal (top) or ventral (bottom) E14 spinal cord NPC grafts at six weeks following transplantation into sites of C4 dorsal column lesion. Immunoreactivity is shown for ventral interneuron marker Foxp2 (left) and dorsal interneuron markers Lbx1 (middle) and Tlx3 (right). **d** Representative images of CGRP[+] axon (cyan) innervation of whole, dorsal, or ventral spinal cord NPC grafts. Graft boundaries indicated with dotted lines. **e** Quantification of CGRP[+] axon density within whole ($n = 5$), dorsal ($n = 6$), and ventral ($n = 5$) spinal cord NPC grafts at 6 weeks post-transplantation. Mean ± SEM; *$p < 0.05$, ***$p < 0.001$ by one-way ANOVA followed by Tukey's multiple comparisons test. **f** Representative images of CST axon (green) regeneration into whole, dorsal, or ventral spinal cord NPC grafts at 6 weeks post-transplantation. Graft boundaries indicated with dotted lines. **g** Quantification of CST axon density within whole ($n = 5$), dorsal ($n = 6$), or ventral ($n = 5$) NPC grafts, normalized to total number of anterogradely-labeled axons in the corticospinal tract rostral to the lesion site. Mean ± SEM; *$p < 0.05$ by one way ANOVA followed by Tukey's multiple comparisons test. Scale bars = 250 μm (**f**), 100 μm (**a**, **d**), 50 μm (**c**).

spinal cord to partially recapitulate normal topographical development following placement into a milieu absent of developmental cues. Here, we demonstrate that dissociated spinal cord NPCs transplanted into the lesioned adult spinal cord form organotypic, dorsal horn-like structures containing multiple cell types arranged in a laminar organization partially recapitulating that of the intact dorsal spinal cord laminae. During normal spinal cord development, late-born dIL precursors arise from asymmetric division of single progenitors between E14-E16, migrating together to the developing dorsal horn where they become intermingled in a "salt-and-pepper" fashion. These cells give rise to virtually all neurons comprising the superficial dorsal laminae via specification mechanisms that are independent of dorsoventral patterning or roof plate signaling[44,60,64,65]. It is therefore likely that individual dorsal horn-like domains within NPC grafts emerge from randomly dispersed, mitotic pdIL progenitors. Following transplantation, these cells may progress through normal differentiation and specification events, giving rise to correctly patterned but randomly oriented dorsal horn-like clusters within grafts.

Notably, when these clusters were positioned adjacent to corticospinal axons penetrating grafts, they appeared to block CST regeneration (Fig. 4g). Thus, to enhance the ability of corticospinal and other functionally important axonal systems to encounter their appropriate graft-derived neuronal targets, spatial engineering of grafts may be required. One means of achieving this spatial organization could be compartmentalized grafts wherein specific spinal cord progenitor domains are either extruded in fibrin/thrombin gels by 3D printing[66,67] or placed into bioengineered multi-channel scaffolds[68–73] in such a way that individual channels of scaffolds are loaded with distinct cell types to interface with specific host axonal populations. In this way, grafts could be shaped to more accurately resemble the topography of the intact spinal cord. Instructing graft development in such a manner might enable greater efficiency of relay formation between distinct host and graft systems. Finally, although our results demonstrate that dorsal horn-like domains are functionally innervated by host nociceptive axons, the extent to which these 'islands' of sensory neurons can integrate into the host nervous system and modulate functional outcomes is an important topic for future study. It is also possible that the formation of ectopic sensory foci may result in adverse outcomes.

There is still much that remains to be investigated regarding the nature of differentiated NPC grafts, including the complete characterization of distinct neural subtypes populating these grafts, as well as the array of complex relay circuits that might spontaneously arise between graft-derived neurons and host sensory and motor systems. We have demonstrated that after transplantation into the injured adult CNS, spinal cord NPCs maintain the capacity to accurately recapitulate normal developmental differentiation and specification programs, to partially recapitulate the assembly of normal dorsal spinal cord cytoarchitecture, and to receive biologically appropriate innervation from regenerating host axon projections. In conjunction with our recent findings that regenerating CST axons form functional synaptic connections with graft-derived neurons that facilitate recovery of fine motor function[7], it appears likely that injured adult corticospinal, and perhaps other populations of host neurons, are intrinsically programmed to spontaneously seek out and synapse onto their normal target neuron subtypes within grafts. Gaining insight into how organization of heterogeneous graft tissue affects growth and connectivity of regenerating host projections will be a critical factor in engineering neural grafts optimized for the assembly of new circuitries to more substantially improve functional outcomes.

## Methods

**Animals**. A total of 119 rats were used for this study, including 77 adult, female Fischer 344 (F-344) rats (150–200 g; Harlan); a total of 9 postnatal male and female rats (age P5, $n = 4$; age P21, $n = 5$); a total of 73 E14 male and female embryonic F-344 rats, including wild-type embryos ($n = 66$) or transgenic F-344 embryos expressing GFP under the ubiquitin promoter ($n = 7$). National Institutes of Health guidelines for laboratory animal care and safety were strictly followed. All animal procedures were approved by the Institutional Animal Care and Use Committee of the Department of Veterans Affairs (VA) San Diego Healthcare System. Animals had free access to food and water throughout the study.

**Cell preparation**. Rat spinal cord neural progenitor cells (NPCs) were prepared as previously described[26]. Spinal cords from E14 rat embryos were dissected in ice-cold HBSS on the morning of grafting. For transplantation of NPCs from dorsal- and ventral embryonic spinal cord tissue (Fig. 5), the dorsal and ventral halves of freshly harvested spinal cords were separated using a #15 scalpel blade. Spinal cord tissue was digested in 0.125% trypsin, dissociated into a single-cell suspension in Neurobasal medium (Life Technologies, Grand Island, New York) + 2% B27 (Life Technologies), and kept on ice prior to in vitro or in vivo experiments. Cell viability was assessed by trypan blue exclusion (Life Technologies); cell viability was always > 90%.

**In vitro NPC characterization**. Immediately following dissociation of spinal cord NPCs, $2 \times 10^6$ viable cells/well were plated onto 48-well tissue culture plates coated with poly-D-lysine (Sigma-Aldrich). Cells were maintained with Neurobasal + 2% B27 + 1% penicillin/streptomycin (Life Technologies) with fresh media changes daily. At either 24 h or 10 d after plating, cells were fixed with 2% paraformaldehyde for 20 min at room temperature and washed with TBS. Fixed cells were blocked for 1 h in TBS + 0.25% Triton-X-100 (TBST) + 5% normal donkey serum, incubated with primary antibodies (see Supplementary Table 1) for 1 h, washed in TBS, incubated with AlexaFluor-conjugated secondary antibodies diluted in blocking buffer (1:500, Thermo Fisher Scientific) for 2 h at room temperature, and washed again with TBS containing DAPI (200 ng/mL) for nuclear staining.

**Spinal cord injury and cell transplantation**. The dorsal column lesion model of spinal cord injury was utilized for this study because it axotomizes 98% of corticospinal axons in the rodent[74] and also axotomizes sensory afferent axons in the dorsal spinal cord. Dorsal column lesions were performed at spinal cord cervical level 4 (C4), as previously described[7]. All surgeries were performed under deep anesthesia using a combination of ketamine (50 mg/kg), xylazine (2.6 mg/kg), and acepromazine (0.50 mg/kg). Following laminectomy, a retracted tungsten wire knife (McHugh Milieux, Downers Grove, IL) was placed 0.6 mm lateral to the midline and inserted to a depth of 1 mm below the dorsal spinal cord surface, then the arc of the knife was extruded 1.5 mm and raised to transect the dorsal columns. Freshly isolated NPCs were washed and resuspended to a concentration of 400,000 viable cells/μL in ice-cold HBSS. Cells were grafted into sites of SCI within 30 min following lesion placement. A 3 μL volume containing $1.2 \times 10^6$ cells was grafted into each lesion cavity using a pulled glass micropipette connected to a PicoSpritzer II (General Valve, Inc., Fairfield, NJ). For dorsal/ventral comparison study, acutely injured subjects were randomly assigned to receive dorsal-($n = 12$), ventral-($n = 10$), or whole ($n = 9$) spinal cord NPC grafts. For all other graft studies ($n = 19$), whole E14 spinal cord grafts were used. The grafted cells were not prepared with a matrix or growth factor cocktail, as NPCs transplanted into a dorsal column lesion site exhibit excellent survival in the absence of these factors, as previously reported[2].

**Capsaicin administration**. Capsaicin (Sigma-Aldrich) was resuspended to a concentration of 2.5% $w/v$ in vehicle (1.5% Tween-80 + 1.5% ethanol in saline) and a total of 250 μL was injected subcutaneously into 6 points along the dorsal aspect of both forelimbs of anesthetized animals using an insulin syringe. Injections were administered to intact, uninjured animals ($n = 3$ capsaicin, $n = 3$ vehicle), as well as to animals at 6 weeks post-SCI + NPC graft ($n = 3$ capsaicin, $n = 3$ vehicle). Animals were sacrificed 90 min following injections.

**Anterograde tracing**. To label corticospinal axons in rats, AAV8 viral vectors (Salk Institute Viral Vector Core, La Jolla, CA) expressing membrane-targeted EGFP or TdTomato transgenes under control of the CAG promoter were injected bilaterally into forelimb and hindlimb areas of primary motor cortex and primary somatosensory cortex (coordinates from bregma: A/P + 1.2 to −1.8 mm; M/L ± 1.7 to 3.7 mm; depth 1.2 to 1.5 mm). For intact controls ($n = 5$), 5 μL of virus at a concentration of $1 \times 10^{13}$ genome copies/mL was injected over 4 sites per hemisphere in age P21 rats. For six-week survival time studies ($n = 46$), 10 μL of virus at a concentration of $1 \times 10^{13}$ genome copies/mL was injected over 8 sites per hemisphere at 3 weeks following SCI and graft. For two-week survival time studies ($n = 4$), biotinylated dextran amine (BDA; MW 10,000 kDa; Molecular Probes) was resuspended to a concentration of 10% $w/v$ in PBS and injected in a volume of 5 μL over 8 sites per hemisphere. Animals were sacrificed at 2–3 weeks following AAV or BDA injection.

**Immunohistochemistry**. All animals were deeply anesthetized and sacrificed by transcardial perfusion with ice-cold saline followed by 4% paraformaldehyde in PBS. Following perfusion, spinal cords were removed and post-fixed overnight in 4% paraformaldehyde, then cryopreserved in a solution of 30% sucrose in 0.1 M phosphate buffer for 2–3 days. Spinal cord tissue was sectioned in either the sagittal or the transverse plane on a sliding microtome (30-μm thickness) and stored as a 1:6 series. For all immunostaining except for detection of CST axons (see below), sections were washed in TBS, blocked for 60 minutes in 5% normal donkey serum in TBST, and incubated with primary antibodies (for list of all antibodies used, see Supplementary Table 1) diluted in blocking buffer at 4 °C overnight. The next day, sections were washed in TBS, incubated with AlexaFluor-conjugated secondary antibodies diluted in blocking buffer for 2 h at room temperature, and washed in TBS. For nuclear staining, DAPI was added to the final wash. Sections were mounted onto gelatin-coated glass microscope slides and coverslipped with Mowiol mounting medium. For enhanced visualization of CGRP$^+$ axons, an antigen retrieval step was included prior to blocking: Following TBS washes, sections were incubated at 80 °C for 30 min in 10 mM sodium citrate buffer, pH 6.0, washed again 3 times in TBS, then blocked and stained as described above.

A biotinylated tyramide amplification protocol[75] was used to boost detection of fluorescently-labeled corticospinal axons. Sections were washed in TBS, incubated for 20 min in TBST, quenched for 30 min in TBS + 0.6% H$_2$O$_2$, and blocked for 60 min in TBST + 5% normal donkey serum. Sections were then incubated overnight with primary antibodies (either anti-GFP or anti-mCherry) diluted in blocking buffer. Sections were washed in TBS and then incubated in biotinylated donkey secondary antibodies (1:300; Jackson Immunoresearch; Cat# 703-065-155, RRID: AB_2313596; Cat# 705-065-003, RRID: AB_2340396; Cat# 711-065-152, RRID: AB_2340593) for 2 h at room temperature. Sections were washed, incubated with VectaStain ABC solution (Vector Laboratories; Cat# PK-6100 RRID: AB_2336819) for 30 min, washed, and then incubated with biotinyl tyramide (1:2500) for 30 minutes. Sections were washed extensively and incubated overnight at 4 °C with AlexaFluor-conjugated streptavidin (1:800; Thermo Fisher Scientific; Cat# S-11226, RRID: AB_2315774; Cat# S11223, RRID: AB_2336881). Finally, sections were washed and mounted as described above.

**Image acquisition, quantification, and statistical analysis**. In vitro NPC cultures were imaged using an ImageXpress Micro system with MetaXpress image acquisition software (Molecular Devices). Tissue sections were imaged using either an Olympus upright fluorescent BX53 microscope equipped with a digital camera, an Olympus FV1000 confocal microscope, or a Keyence BZ-X700 fluorescent microscope. Experimenters were blinded to experimental groups during image processing and quantification. Image quantification was performed using ImageJ software, and statistical analysis was performed with GraphPad Prism 6.0 software; $p$ values less than 0.05 were considered statistically significant. NPC graft boundaries were determined by NeuN, DAPI, and/or GFP immunoreactivity. For all quantitative and qualitative comparisons, NPC grafts were age-matched to control spinal cord tissue from uninjured (intact) rats (2-week-old grafts matched to intact cord of age P5 rats; 6-week-old grafts matched to intact cord of age P35 rats).

For quantification of cellular markers in NPC cultures (Fig. 1), three technical replicates (cultures of cells obtained from three separate E14 spinal cord dissociations) were performed for both 24 h and 10 day time-points. MetaXpress images were automatically thresholded in ImageJ, and numbers of cells immunoreactive for NPC and neuronal markers were automatically quantified and normalized to total numbers of DAPI$^+$ nuclei.

Quantification of host CGRP$^+$ and anterogradely-traced CST axon density was performed by automatically thresholding axon signal intensity in ImageJ. For quantification of CGRP$^+$ and CST axon density within regions of dorsal horn calretinin$^+$ immunoreactivity in intact spinal cord and NPC grafts (Supplementary Fig. 4), regions encompassing calretinin$^+$ clusters were drawn automatically using signal thresholding. Host axon intensity was thresholded, and pixel density within calretinin$^+$ regions was normalized to total gray matter/graft pixel density. CST axon density within grafts was normalized to the number of anterogradely-labeled main tract axons in the dorsal CST in transverse sections 5 mm rostral to the lesion site. Avoidance/enrichment data were log$_2$-transformed and analyzed by single-sample $t$-test compared to the hypothetical value of 0 (no change). One section per animal was quantified for intact spinal cord; 2–7 sections per animal were quantified for NPC grafts. For quantification of percent CST fiber density by laminae in intact, transverse spinal cord sections (Fig. 4c), boundaries of spinal cord laminae were determined based on NeuN staining and rat spinal cord atlas images were consulted as a reference. CST axon intensity was thresholded, and pixel density in each ROI was normalized to CST pixel density within entire spinal cord gray matter for each image. One spinal cord section per animal was quantified.

For quantification of NeuN$^+$ neurons and nuclear markers (Foxp2, Tlx3, Lbx1) in NPC grafts (Supplementary Fig. 6c–e), nuclear signal intensity was thresholded and counted automatically using ImageJ, and number of nuclei were normalized to the volume of each graft section (ROI area × 30 μm thickness). Percent graft volume occupied by calretinin$^+$ clusters (Supplementary Fig. 6f) was calculated by automatic thresholding of calretinin$^+$ regions. Four to nine sections per animal were quantified. Quantification of graft volume was determined by summing the area of every 6th sagittal section through grafts and mean graft volume was

calculated for each group. For quantification of c-Fos$^+$ nuclei in grafts (Fig. 3), ROIs were drawn around calretinin$^+$ regions as described above. For each image, each calretinin$^+$ region was visually inspected and categorized as "with" or "without" CGRP$^+$ axon innervation. A ROI was classified as "with" CGRP$^+$ axon innervation if at least one CGRP$^+$ axon was present within the ROI. c-Fos$^+$ nuclei were quantified using automatic signal thresholding and quantification in ImageJ.

**Data availability**. The data that support the findings of this study are available from the corresponding author upon reasonable request.

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

## Acknowledgements

We thank L. Graham, E. Staufenberg, S. Im, and A. Rizzo for their technical assistance, and C. Birchmeier and T. Müller for the generous gifts of Tlx3 and Lbx1 antibodies. This work was supported by the Craig H. Neilsen Foundation, the US Veterans Administration Gordon Mansfield Spinal Cord Injury Consortium, the National Institutes of Health (NS042291), and the Dr. Miriam and Sheldon G. Adelson Medical Research Foundation.

## Author contributions

J.N.D., A.F.A., C.L-K., J.W.F. and M.H.T. designed the study. J.N.D. and M.H.T. wrote the manuscript. J.N.D., A.F.A., H.K., C.L-K., G.H.D.P., H.S., D.G., K.K. and P.L. performed experiments. J.N.D. and A.F.A. analyzed data.

## Additional information

**Competing interests:** The authors declare no competing financial interests.

