## [Peer review file · Nature Communications]

Reviewers' comments:

Reviewer #1 (Remarks to the Author):

This manuscript addresses two interesting and important questions in the field transplantation of neural progenitors (NPCs) into the central nervous system (CNS) for repair purposes. First, to what degree might NPCs reform an organized structure after grafting, and second, to what degree might NPCs receive appropriate and cell-type specific innervation from host axons entering the graft. The findings presented provide new and compelling proof-of-principle evidence for both of these points. In this study, the authors convincingly show that embryonic spinal cord NPCs grafted into the injured spinal cord spontaneously organized into multicellular 'organotypic' structures that exhibited the expected laminar organization of spinal cord dorsal horn. Moreover, these graft-derived dorsal-horn-like structures received axonal contacts from appropriate host ascending sensory axons, and were avoided by host descending corticospinal tract 'motor' axons.

Overall, I found that the study was well conducted and appropriately controlled. The data look to be of very high quality. The findings presented convincingly, and in many cases beautifully, support the interpretations made in the text. I have no major comments or concerns regarding the data presented or their interpretation. Nevertheless, there are a few points that should be dealt with as mentioned below.

One thing I paid particular attention too in reading this manuscript was to assess what was new in this paper. This laboratory has over the past few years published a fair number of papers in high profile journals on the general topic of grafting of embryonic NPC after spinal cord injury. So I felt it important to determine if a new major point was being made. As regards this particular manuscript, the answer is clearly and emphatically yes. This study poses two questions that are important for understanding the biology of NPC grafting in the CNS and provides compelling new information relevant to those questions. Only in this manner will the potential of NPC grafting be realized. In this way, this study stands out in this field and makes an important new contribution that represents a clear advance in the field. The notion that grafted NPC can self-organize into appropriate multicellular structures and receive appropriate host innervation while avoiding inappropriate innervation are major new points. I think the study is appropriate for, and will be of interest to, a broad audience.

Specific comments and concerns:

(line numbers refer to those on the left of the pdf of the text)

(1) In the abstract (line 32) the word "laminated" should be changed to "laminar". Per various dictionaries, laminated means 'coated with plastic' and laminar means 'consisting of laminae', which is what is intended here.

(2) Without being excessively picky about grammar, the word "undergo" in line 53 in the Introduction (and again in line 267 of the Discussion) confused me and I had to read the sentence several times before I understood what was meant. In dictionaries, undergo means 'to be subjected to' or 'to receive' and so my first interpretation of these sentences was that the authors were saying that there is considerable published evidence that grafted neurons receive specific innervation. Interpretation of this statement in that manner, however, goes counter to one of the main points the authors are trying to make. It took me a bit to understand that the word 'undergo' was being misused. The authors should change 'undergo' to 'send' or 'project', or in some other way edit this sentence to be more clear.

Reviewer #2 (Remarks to the Author):

This is a very interesting manuscript providing evidence that when transplanted into rat spinal cord

injury embryonic rat spinal cord neural progenitor cells form laminated structures probably according to their identity of origin, and display target attraction/repulsion properties appropriate for their identity. These laminated structures seem to be dispersed within the transplanted spinal cord site. This ability to organized into laminated structure points to the future possibility of engineering spatially appropriate regions of spinal cord from embryonic neural progenitor cells.

This is certainly very interesting work that points in an interesting direction but requires some further characterization before being suitable for publication Nature Communications.

1. Figure 1A shows an example of the GFP cells in the graft, but in all other figures, the GFP expression is not shown. Therefore, in the laminated structures, it cannot be assessed if the laminated structures are composed entirely of GFP cells, or whether host cells have been recruited into the region. More complete documentation of the graft versus host cells would be important.
2. Given the past and also quite recent work indicating that transplanted neural precursor cells can fuse with host cells in damage sites, it would be important to establish if the transplanted GFP cells are fusing with host cells to establish identity. Transplantation into an RFP expressing host would be appropriate.
3. It would be good to know which of these markers used for analysis are already expressed in the rat E14 NPC preps prior to transplantation. Although this does not speak to whether some NPC cells might turn on the marker after transplantation, it would be good to provide an idea of what kinds of cells are going into the transplant. There is scant information on the cell preparation.
4. The authors carefully quantitate the number of cells that are being transplanted. It would be good to provide some estimates of how many cells are remaining in the transplant.

Reviewer #3 (Remarks to the Author):

The manuscript presents novel and interesting findings describing the fate and integrative capabilities of neural progenitor cell (NPC) transplantation in the injured spinal cord utilizing sets of location-specific neuronal markers in conjunction with axonal tracing. The findings indicate that NPC transplants derived from E14 spinal cords can self-assemble into laminated clusters with appropriate dorsal horn-like domains recapitulating the intact age-matched neuron-specific lamination pattern in the spinal cord dorsal horn. The photomicrographs showing this laminar arrangement in clusters within the NPC grafts using markers for subpopulations of lamina I/II neurons (Tlx3, calretinin, calbindin) and lamina III/IV neurons (Lbx), some even in arc shapes similar to dorsal horn, are particularly intriguing. In addition, host axon regeneration patterns after injury appear to show appropriate potential reinnervation with sensory (CGRP) fibers while blocking reinnervation by inappropriate corticospinal tract axons. These findings were further confirmed using dorsal or ventrally derived NPCs which were enriched in spatially expected neuronal phenotypes and innervation. Together, the data suggest that functionally relevant host-graft integration may be achievable using spatially oriented site specific NPC grafts.

There are some minor concerns:

Although host CGRP axon terminals with synaptophysin appeared in proximity with calretinin neuron clusters in NPC grafts, it is difficult to discern actual presynaptic contacts in the photomicrographs at this level (e.g. Fig. 2e); thus suggesting synaptic connectivity may be premature in the absence of confirmation by electron microscopy. There are also some calretinin clusters in the NPC grafts that do not appear to receive any CGRP innervation at all (e.g. lower left clusters in Fig. 2d).

It is unclear why a similar analysis of potential innervation by corticospinal tract axons was not done for the Lbx subpopulation of the clusters since the laminar home of this phenotype appears

to be III-IV which receives corticospinal input (e.g. Fig. S3). Thus it would seem like an important part of the puzzle to support the hypothesis that adult host axons retain their ability after injury to recognize appropriate targets within the grafts.

Since there are multiple clusters of mini dorsal horn-like arrangements scattered throughout the NPC grafts, relevant functional integration and communication with host neuronal circuitry may not be possible even if the grafted cells successfully receive synaptic connections from the correct host axonal origin. In addition, this may result in foci of abnormal hyperexcitability or undesirable effects such as pain (particularly with increased CGRP innervation to grafted excitatory NPC neurons like Tlx3 subtype). A means to achieve spatial organization, as suggested in the discussion, will likely be necessary, as well as a means to inhibit the formation of abnormal foci.

Reviewer #1:

Critique:

(1) In the abstract (line 32) the word “laminated” should be changed to “laminar”. Per various dictionaries, laminated means ‘coated with plastic’ and laminar means ‘consisting of laminae’, which is what is intended here.

Response: Done. Thank you for this information.

(2) Without being excessively picky about grammar, the word “undergo” in line 53 in the Introduction (and again in line 267 of the Discussion) confused me and I had to read the sentence several times before I understood what was meant. In dictionaries, undergo means ‘to be subjected to’ or ‘to receive’ and so my first interpretation of these sentences was that the authors were saying that there is considerable published evidence that grafted neurons receive specific innervation. Interpretation of this statement in that manner, however, goes counter to one of the main points the authors are trying to make. It took me a bit to understand that the word ‘undergo’ was being misused. The authors should change ‘undergo’ to ‘send’ or ‘project’, or in some other way edit this sentence to be more clear.

Response: Thank you again. We now use the term “spontaneously innervate their correct targets” instead of “spontaneously undergo innervation” (lines 53-54, Introduction), and “innervate appropriate subtype-specific target” instead of “undergo subtype-specific innervation” (lines 323-24, Discussion).

Reviewer #2:

Critique 1A: Given the past and also quite recent work indicating that transplanted neural precursor cells can fuse with host cells in damage sites, it would be important to establish if the transplanted GFP cells are fusing with host cells to establish identity. Transplantation into an RFP expressing host would be appropriate.

Critique 1B: Figure 1A shows an example of the GFP cells in the graft, but in all other figures, the GFP expression is not shown. Therefore, in the laminated structures, it cannot be assessed if the laminated structures are composed entirely of GFP cells, or whether host cells have been recruited into the region. More complete documentation of the graft versus host cells would be important.

Response: We thank the reviewer for these insightful comments. The questions of whether cell fusion occurs between graft- and host neurons, or whether the host spinal cord environment might otherwise modulate graft-derived neuronal fates, are important considerations. We now provide extensive additional data illustrating an absence of host-graft neuronal cell fusion, and demonstrating that grafted NPCs establish defined neuronal subtype identities and assemble multicellular laminar structures in the absence of contact and/or fusion with host neurons.

First, regarding the issue of whether cell-cell fusion occurs between host and graft in this model: we have previously shown that host/graft cell fusion does not occur within transplants of RFP⁺ human iPSC-NSCs into GFP⁺ host rats (Fig. 4 from Lu *et al.*, Neuron, 2014). [Redacted].

[Redacted]

[Redacted]

[Redacted]

For the current study, we have now added additional confirmation of graft-derived neuronal identity, listed below, that comprise Figure 1 & Supplementary Figure 3c in the revised manuscript, as follows:

Figure 1: We now provide data illustrating the acquisition of dorsal spinal cord neuronal identities by NPCs cultured *in vitro*. We find that all of the dorsal horn interneuron subtypes that comprise dorsal horn-like domains in mature grafts, are also present in NPCs cultured for 10 days (**Fig. 1f-l**). Moreover, we show that these dorsally-fated neurons are located within discrete clusters in culture, similar to the clusters formed in grafts *in vivo*. These data provide a clear demonstration that the acquisition of fates/identities of NPC-derived neurons are *intrinsic* properties of these cells, because these fates are adopted in the absence of a host environment.

Suppl. Figure 3c: For additional confirmation that these structures are not comprised of host neurons, we now show that all neurons within these domains are GFP⁺ (**Suppl. Fig. 3c**). These images clearly demonstrate GFP expression of calretinin⁺ (laminae I/II) and Lbx1⁺ (laminae III/IV) by neurons within GFP⁺ graft tissue, unequivocally indicating graft-derived identity. These data demonstrate the ability of donor NPCs to assemble multicellular dorsal horn-like domains following transplantation into the uninjured *in vivo* host white matter environment.

Together, these new data demonstrate that laminar structures arising from NPC grafts are graft-derived. Moreover, these data highlight the ability of NPCs to differentiate into specific dorsal and ventral neuronal identities when placed into a lesion site.

Critique 2: It would be good to know which of these markers used for analysis are already expressed in the rat E14 NPC preps prior to transplantation. Although this does not speak to whether some NPC cells might turn on the marker after transplantation, it would be good to provide an idea of what kinds of cells are going into the transplant. There is scant information on the cell preparation.

Response: Our characterization of graft-derived neuronal subtypes in the present study is dependent upon expression of terminal differentiation markers in specific neuronal classes or regional domains within the *postnatal/adult* spinal cord. For example, Tlx3 and calretinin are expressed by mature laminae I/II neurons, and Foxp2 is expressed by mature V1 neurons in the ventral cord. These markers appear at postnatal ages (weeks after grafting). Hence, we cannot use these same markers to characterize specific progenitor types present in the E14 NPC preparation used for grafting. To address the reviewer's question, we have characterized the expression of a select number of developmental transcription factors in E14 cells after culturing for 24 h *in vitro* (**Fig. 1a-e**). By identifying major dorsal/ventral NPC populations in these cells, we now show that multiple progenitor types are present in the cell preparation used for grafting. Although this does not speak to the relative survival/proliferation capacities of distinct progenitor types after grafting into the lesioned adult CNS environment, we hope that the current data will provide a sufficient degree of detail about the types of progenitors present in our cell preparation used for grafting.

Critique 3: The authors carefully quantitate the number of cells that are being transplanted. It would be good to provide some estimates of how many cells are remaining in the transplant.

Response: We have now quantified the number of NeuN⁺ neurons present in mature NPC grafts at 6 weeks post-transplantation and report the Results on page 5 of the revised manuscript. A mean of $34,900 \pm 1540$ (\pm SEM) NeuN⁺ cells/mm³ of graft tissue were present ($n = 9$ animals).

Reviewer #3 (Remarks to the Author):

The manuscript presents novel and interesting findings describing the fate and integrative capabilities of neural progenitor cell (NPC) transplantation in the injured spinal cord utilizing sets of location-specific neuronal markers in conjunction with axonal tracing. The findings indicate that NPC transplants derived from E14 spinal cords can self-assemble into laminated clusters with appropriate dorsal horn-like domains recapitulating the intact age-matched neuron-specific lamination pattern in the spinal cord dorsal horn. The photomicrographs showing this laminar arrangement in clusters within the NPC grafts using markers for subpopulations of lamina I/II neurons (Tlx3, calretinin, calbindin) and lamina III/IV neurons (Lbx), some even in arc shapes similar to dorsal horn, are particularly intriguing. In addition, host axon regeneration patterns after injury appear to show appropriate potential reinnervation with sensory (CGRP) fibers while blocking reinnervation by inappropriate corticospinal tract axons. These findings were further confirmed using dorsal or ventrally derived NPCs which were enriched in spatially expected neuronal phenotypes and innervation. Together, the data suggest that functionally relevant host-graft integration may be achievable using spatially oriented site specific NPC grafts.

There are some minor concerns:

Critique 1: Although host CGRP axon terminals with synaptophysin appeared in proximity with calretinin neuron clusters in NPC grafts, it is difficult to discern actual presynaptic contacts in the photomicrographs at this level (e.g. Fig. 2e); thus suggesting synaptic connectivity may be premature in the absence of confirmation by electron microscopy.

Response: First, we have performed a new experiment (**Fig. 3d-h**) demonstrating the functional transmission of sensory information from host fibers to appropriately innervated domains in graft. Following subcutaneous injection of the TRPV1 agonist capsaicin, we have found that calretinin⁺ neuron clusters that are innervated by CGRP⁺ host fibers—but *not* clusters that are *not* innervated by CGRP⁺ host fibers—exhibit robust c-Fos expression in calretinin⁺ neuron clusters. This result indicates that functionally appropriate graft neuronal targets become activated secondary to peripheral noxious stimulation.

Second, since the initial review of the current manuscript, our group has published a separate report describing a comprehensive characterization of monosynaptic host inputs onto NPC grafts placed in sites of SCI, using the modified rabies virus system. In that study, we demonstrated that host CGRP⁺ sensory neurons (as well as many other host spinal and supraspinal neuronal populations) form functional synapses onto graft-derived targets (Fig. 4 in Adler *et al.*, Stem Cell Reports, 2017). The Adler paper confirms synaptic connectivity between CGRP⁺ afferents and NPC graft-derived neurons, although the report did not investigate target specificity of host-to-graft innervation. We cite this work in the revised manuscript as confirmation of CGRP⁺ host axon synaptic connectivity with grafts.

Together, these two points suggest functional synaptic connectivity between host sensory axons and their graft-derived targets.

Critique 2: There are also some calretinin clusters in the NPC grafts that do not appear to receive any CGRP innervation at all (e.g. lower left clusters in Fig. 2d).

Response: The review is correct. Calretinin clusters located in closer proximity to incoming sensory axons from DRGs (that is, located in dorsal regions of the graft) are innervated, whereas more distant (rostrally located) calretinin clusters are sparsely innervated. We have added this information to the revised manuscript (page 7).

Critique 3: It is unclear why a similar analysis of potential innervation by corticospinal tract axons was not done for the Lbx subpopulation of the clusters since the laminar home of this phenotype appears to be III-IV which receives corticospinal input (e.g. Fig. S3). Thus it would seem like an important part of the puzzle to support the hypothesis that adult host axons retain their ability after injury to recognize appropriate targets within the grafts.

Response: We performed this quantification and found that in NPC grafts, CST axons avoided Lbx1⁺ neuron clusters, albeit to a lesser degree than they avoided calretinin clusters. However, in contrast to other axonal systems whose post-synaptic targets are well characterized, corticospinal axons have a rather broad pattern of termination in spinal cord gray matter and have a variety of targets, many of which have not yet been characterized. Future work is needed to identify specific cellular targets of CST axons in the *intact* system, so that host-graft connectivity for corticospinal axons can be better characterized in the injured system. We are performing such studies currently using anterograde trans-synaptic tracers.

Critique 4: Since there are multiple clusters of mini dorsal horn-like arrangements scattered throughout the NPC grafts, relevant functional integration and communication with host neuronal circuitry may not be possible even if the grafted cells successfully receive synaptic connections from the correct host axonal origin. In addition, this may result in foci of abnormal hyperexcitability or undesirable effects such as pain (particularly with increased CGRP innervation to grafted excitatory NPC neurons like Tlx3 subtype). A means to achieve spatial organization, as suggested in the discussion, will likely be necessary, as well as a means to inhibit the formation of abnormal foci.

Response: We agree. New methods to achieve spatial organization of grafts will be an important next step toward engineering neural grafts that are more functionally 'streamlined' to promote maximal modality-specific connectivity. While engineering methods for achieving such organization is beyond the scope of this paper, this is a key area of ongoing investigation in our group.

Once again, we thank the reviewers for their insightful comments and we hope that these responses satisfactorily address their concerns.

Sincerely yours,

Mark H. Tuszynski, M.D., Ph.D.

References cited:

- Adler AF, Lee-Kubli C, Kumamaru H, Kadoya K, Tuszynski MH (2017). Comprehensive monosynaptic rabies virus mapping of host connectivity with neural progenitor grafts after spinal cord injury. *Stem Cell Reports* 8(6): 1525-33.
- Lu P, Woodruff G, Wang Y, Graham L, Hunt M, Wu D, Boehle E, Ahmad R, Poplawski G, Brock J, Goldstein LS, Tuszynski MH (2014). Long-distance axonal growth from human induced pluripotent stem cells after spinal cord injury. *Neuron* 83(4): 789-796.

REVIEWERS' COMMENTS:

Reviewer #1 (Remarks to the Author):

In this revised version, the authors have appropriately dealt with my previous comments concerns. In my opinion, they have also appropriately dealt with the concerns of the other referees as well. They have added new data of good quality, and I think the paper is strong and makes two important points that merit broad exposure, as outlined in my first review.

Reviewer #2 (Remarks to the Author):

The authors have taken extensive efforts to address the reviewers' comments to my satisfaction.

Reviewer #3 (Remarks to the Author):

The revised manuscript has sufficiently addressed the main points of the previous critique. In particular, the use of capsaicin to induce c-Fos activation adds to the interpretation of functional synaptic integration of grafted NPCs. It is somewhat unclear how the borderline between the CGRP+ and CGRP- regions were delineated for the separated c-Fos quantification (Fig. 3), since it appears that c-Fos labeling is distributed fairly evenly throughout the graft. The CGRP innervation is also not as crisp (blue in Fig. 3f-g) compared with other examples (Fig. 3c, Fig. 5d), thus the boundaries between CGRP-innervated and non-innervated graft regions appear difficult to discern for the quantitative comparisons. Perhaps this can be delineated in the Figs. The observation that there was no significant difference in total numbers of c-Fos labeling between capsaicin and vehicle treated animals when all regions of the graft tissue is included is puzzling since it seems as though overall c-Fos is increased substantially in response to noxious stimulation.

There is still some concern that the clusters of NPC grafts are functioning as independent foci and may not integrate in a meaningful way within the host neurocircuitry. While the authors suggest future manipulations to address this (e.g. spatial engineering of the grafts with 3D gels), at this stage the findings, while interesting, are somewhat limited. In addition, possible undesired outcomes, such as generation of abnormal ectopic foci may result. This does not detract from the current findings, but caveats and limitations should be included in the discussion.

Reviewer #3:

Critique: The revised manuscript has sufficiently addressed the main points of the previous critique. In particular, the use of capsaicin to induce c-Fos activation adds to the interpretation of functional synaptic integration of grafted NPCs. It is somewhat unclear how the borderline between the CGRP+ and CGRP- regions were delineated for the separated c-Fos quantification (Fig. 3), since it appears that c-Fos labeling is distributed fairly evenly throughout the graft. The CGRP innervation is also not as crisp (blue in Fig. 3f-g) compared with other examples (Fig. 3c, Fig. 5d), thus the boundaries between CGRP-innervated and non-innervated graft regions appear difficult to discern for the quantitative comparisons. Perhaps this can be delineated in the Figs. The observation that there was no significant difference in total numbers of c-Fos labeling between capsaicin and vehicle treated animals when all regions of the graft tissue is included is puzzling since it seems as though overall c-Fos is increased substantially in response to noxious stimulation.

Response: A description of how calretinin neuron domains were classified as CGRP⁺ vs. CGRP⁻ was included in the last paragraph of the Methods section (lines 603-605): “*For each image, each calretinin⁺ region was visually inspected and categorized as ‘with’ or ‘without’ CGRP⁺ axon innervation. A ROI was classified as ‘with’ CGRP⁺ axon innervation if at least one CGRP⁺ axon was present within the ROI.*”

Although the overall numbers of c-Fos⁺ nuclei in grafts were not significantly different between vehicle and capsaicin treatment, the relevant comparison is the effect of vehicle vs. capsaicin treatment on CGRP⁺ axon-innervated calretinin⁺ domains in grafts. Because other regions of grafts are not innervated by host C-fibers, c-Fos⁺ immunoreactivity in this tissue may be a result of nonspecific neuronal activity that is independent of the noxious stimulation paradigm employed here. We quantified c-Fos⁺ immunoreactivity specifically within calretinin neuron clusters in order to take advantage of the preferential innervation of host nociceptive fibers for their functionally appropriate target tissue and delineate the effect of noxious stimulation.

Critique: There is still some concern that the clusters of NPC grafts are functioning as independent foci and may not integrate in a meaningful way within the host neurocircuitry. While the authors suggest future manipulations to address this (e.g. spatial engineering of the grafts with 3D gels), at this stage the findings, while interesting, are somewhat limited. In addition, possible undesired outcomes, such as generation of abnormal ectopic foci may result. This does not detract from the current findings, but caveats and limitations should be included in the discussion.

Response: We agree that is difficult to say what would constitute “meaningful” integration at this early stage of investigation. Based on the current body of results in conjunction with recently published rabies tracing experiments in Adler *et al.* (Stem Cell Reports, 2017), host nociceptive axons clearly make functional synaptic connections onto their phenotypically appropriate targets in grafts. Future work will explore the extent to which graft and host systems may integrate, and whether they can restore lost sensory function. We have addressed the reviewer’s concerns in the revised Discussion.

Once again, we thank the reviewers for their insightful comments and we hope that these responses satisfactorily address their concerns.